# Impact of Pre-Pregnancy Body Mass Index on Pregnancy and Perinatal Outcomes in Liver Transplant Recipients: A Retrospective Cohort Study

**DOI:** 10.3390/diagnostics15162054

**Published:** 2025-08-16

**Authors:** Eliza Kobryn, Zoulikha Jabiry-Zieniewicz, Nicole Akpang, Krzysztof Zieniewicz, Michal Grat, Artur Ludwin, Monika Szpotanska-Sikorska

**Affiliations:** 11st Department of Obstetrics and Gynecology, Medical University of Warsaw, 02-015 Warsaw, Poland; eliza.kobryn@gmail.com (E.K.); nakpang2002@gmail.com (N.A.);; 2Department of General, Transplant and Liver Surgery, Medical University of Warsaw, 02-097 Warsaw, Poland; krzysztof.zieniewicz@wum.edu.pl (K.Z.);

**Keywords:** liver transplantation, pregnancy, maternal outcomes, body-mass index, gestational complications, immunosuppression, fetal well-being

## Abstract

**Background**: Pre-pregnancy overweight and obesity are established risk factors for adverse maternal and perinatal outcomes in the general obstetric population. However, data regarding their impact in female liver transplant recipients remain limited. This study aimed to evaluate the association between pre-pregnancy body mass index (BMI) and pregnancy-related complications and neonatal outcomes in this high-risk cohort. **Methods**: A retrospective cohort analysis was conducted on pregnancies in liver transplant recipients who delivered between 2001 and 2022 at a single tertiary referral center. Participants were stratified into two groups based on pre-pregnancy BMI: normal weight (18.5–24.9 kg/m^2^) and overweight/obese (≥25 kg/m^2^). Maternal characteristics, pregnancy complications, and perinatal outcomes were compared using appropriate statistical methods, with significance set at *p* < 0.05. **Results**: Among 72 pregnancies included in the analysis, 48 (66.7%) were in women with normal BMI, and 24 (33.3%) were in those with an elevated BMI. No statistically significant differences were observed in gestational age at delivery, neonatal birth weight, Apgar scores, or incidence of preterm birth. Although pregnancy-induced hypertension and cesarean delivery were more prevalent among overweight/obese individuals, these differences did not reach statistical significance (PIH: 28% vs. 10.4%, *p* = 0.112; cesarean delivery: 76% vs. 64.6%, *p* = 0.465). **Conclusions**: In conclusion, pre-pregnancy overweight and obesity were not significantly associated with adverse obstetric or neonatal outcomes in liver transplant recipients. Nevertheless, the observed trends suggest a potential predisposition to hypertensive disorders (PIH: 28% vs. 10.4%, *p* = 0.112), underscoring the importance of individualized preconception counseling and weight optimization strategies in this high-risk patient population.

## 1. Introduction

The first successful liver transplantation was performed in 1963 [1]. That procedure started a new era in the management of end-stage liver failure. Since then, due to the development of surgical techniques, operator’s experience and the improvement of immunosuppressive therapy, the mean survival period in patients with terminal liver disease has significantly improved [2].

Nowadays, liver transplantation (LT) remains the gold standard for treating end-stage liver disease [3]. Moreover, the restoration of normal liver function contributes to the recovery and functional improvement of other organs too [4]. Amenorrhea and infertility are prevalent among women with chronic liver disease, affecting up to 59% of patients in the year preceding liver transplantation [5]. In women after liver transplantation, the reproductive system undergoes functional restoration within a year post LT, characterized by the normalization of menstrual cycles, the reactivation of the hypothalamic–pituitary–gonadal axis, and the resumption of ovulation. Consequently, many recipients experience a renewed desire for pregnancy and motherhood [5].

Pregnancy after liver transplantation remains a significant challenge for obstetricians and transplant specialists, as its management requires a multidisciplinary approach to balance the risks associated with immunosuppressive therapy, potential graft complications, and maternal–fetal health as, although pregnancies following liver transplantation have been shown to be viable, they are associated with a higher incidence of obstetric complications such as gestational hypertension, preeclampsia, gestational diabetes mellitus, preterm birth or unplanned cesarean sections in comparison to the population [6].

In addition to general maternal risk factors, several transplant-related variables significantly influence obstetric outcomes in liver transplant recipients. The interval between transplantation and conception plays a pivotal role; pregnancies occurring less than 12 months after transplantation are associated with higher risks of cellular rejection, graft dysfunction, and adverse perinatal outcomes such as preterm birth [7]. Therefore, pregnancy is not recommended within the first 24 months after liver transplantation [8]. Despite the increased risks, pregnancies occurring within the first 12 months post-transplant can still result in highly probable favorable outcomes, allowing for cautious optimism [7].

The type and dosage of immunosuppressive therapy are also critical. While agents like tacrolimus and azathioprine are generally considered safe in pregnancy [9], others such as mycophenolate mofetil and mTOR inhibitors are teratogenic and associated with increased risks of spontaneous abortion and congenital anomalies [10,11]. Data on sirolimus use in pregnancy are limited [9]. However, animal studies have demonstrated adverse effects, and when combined with cyclosporine, sirolimus was associated with increased fetal mortality and a higher number of resorptions. As a result, current recommendations advise discontinuing sirolimus prior to conception and transitioning the patient to an alternative, pregnancy-compatible immunosuppressive regimen if needed [9].

Moreover, elevated blood levels of immunosuppressants may contribute to a higher risk of maternal hypertension and renal dysfunction, whereas subtherapeutic levels increase the risk of graft rejection [12].

Pre-pregnancy BMI is increasingly recognized as a key determinant of both maternal and perinatal outcomes in the general obstetric population. Elevated pre-pregnancy BMI is associated with a significantly increased risk of gestational diabetes mellitus, gestational hypertension, and cesarean delivery, as well as neonatal complications such as macrosomia, birth injuries, and difficulties with breastfeeding [13,14,15]. Notably, studies indicate that pre-pregnancy BMI may be a stronger predictor of certain complications—such as gestational diabetes—than gestational weight gain itself [14].

Moreover, maternal overweight and obesity have been correlated with an increased birth weight and length, as well as a higher prevalence of macrosomia [14,16]. Although gestational weight gain had a more pronounced impact on neonatal anthropometric measures, pre-pregnancy BMI was independently associated with adverse perinatal outcomes, including lower Apgar scores and an increased risk of perinatal trauma [14].

Given the established role of BMI in determining pregnancy outcomes, it becomes particularly important to examine how these variables influence pregnancies in women with complex medical backgrounds, such as liver transplant recipients. These patients already face elevated baseline risks due to immunosuppressive therapy and graft-related complications; thus, the presence of additional risk factors like elevated BMI may further complicate maternal–fetal health.

The primary objective of the presented study was to evaluate the association between pre-pregnancy body mass index (BMI) and pregnancy-related complications and neonatal outcomes in women with a history of liver transplantation. By comparing clinical characteristics and perinatal endpoints between liver transplant recipients with normal weight and those who were overweight or obese prior to conception, the study aimed to determine whether elevated BMI constitutes an additional risk factor in this medically complex population. In addition, the study assessed the distribution of body weight and BMI in the liver transplant cohort relative to the general population of pregnant women in Poland, in order to explore potential differences in the anthropometric profiles between transplant recipients and healthy obstetric patients. This comparison provides a broader epidemiological context and may help identify population-specific trends relevant to preconception care.

The principal aim of this study was to evaluate the association between pre-pregnancy body mass index (BMI) and pregnancy-related complications and neonatal outcomes in female liver transplant recipients.

## 2. Materials and Methods

This retrospective, observational cohort study was conducted at the 1st Department of Obstetrics and Gynecology in collaboration with the Department of General, Transplant, and Liver Surgery, Medical University of Warsaw, Poland. Given the non-interventional and retrospective nature of the study, formal approval from the Bioethics Committee was not required; however, the Committee was duly informed regarding the conduct of the research.

Data were retrospectively collected from the medical records of pregnant women with a history of liver transplantation who delivered between 2001 and 2022 at the 1st Department of Obstetrics and Gynecology, Medical University of Warsaw.

The eligibility criteria for inclusion in the study comprised the following: maternal age between 18 and 40 years at the time of delivery, singleton pregnancy, delivery at or beyond 23 weeks of gestation, and history of at least one liver transplantation. The exclusion criteria included the following: multiple gestations, delivery prior to 23 weeks of gestation, maternal age below 18 or above 40 years, history of multiple liver transplantations, multi-organ transplantation (e.g., liver-kidney, liver-pancreas), presence of lethal congenital fetal anomalies and incomplete medical documentation.

Participants were stratified into two subgroups based on pre-pregnancy BMI: normal weight (18.5–24.9 kg/m^2^) and overweight/obese (≥25 kg/m^2^). The following variables were analyzed and compared between these subgroups: primary indication for liver transplantation, type of immunosuppressive regimen, and pre-pregnancy BMI. Additionally, selected perinatal outcomes were evaluated, including gestational age at delivery, neonatal birthweight, 5 min Apgar score and mode of delivery. Each delivery was treated as an independent case and referred to as an ‘event’ in the results section, in order to minimize potential misinterpretations and analytical errors.

Statistical analysis was performed using the Statistica version 14.2.0 program. Categorical variables were compared using the chi-square test or Yates’ corrected chi-square test, as appropriate. Continuous variables were analyzed using Student’s *t*-test for normally distributed data or the Mann–Whitney U test for non-parametric data. A *p*-value of <0.05 was considered statistically significant.

## 3. Results

Between 2001 and 2022, a total of 82 deliveries were recorded in 70 liver transplant recipients at the 1st Department of Obstetrics and Gynecology. Out of this cohort, 52 patients, accounting for 74 deliveries, met the predefined inclusion criteria and were eligible for further evaluation. Initially, participants were categorized into four subgroups based on their pre-pregnancy body mass index (BMI): underweight—2 patients (2.7%), normal weight—48 patients (64.0%), overweight—20 patients (26.62%), and obese—5 patients (6.67%). Due to the limited size of the underweight group, these two cases were excluded from the subsequent statistical analysis. Ultimately, 72 deliveries were included in the final analysis. For statistical purposes, the cases were stratified into two BMI-based subgroups: normal weight (BMI 18.5–24.9 kg/m^2^) and overweight/obese (BMI ≥ 25.0 kg/m^2^). To avoid potential ambiguity and ensure methodological clarity, each delivery was treated as an independent case and, for the purpose of consistency, referred to as an “event” throughout the study.

### 3.1. Baseline Patients Characteristics

The final cohort included 72 events, stratified into two groups based on pre-pregnancy BMI: normal weight (*n* = 48, 66.7%) and overweight/obese (*n* = 24, 33.3%). Maternal age at the time of delivery was comparable between groups, with a mean of 28.96 ± 5.42 years in the normal BMI group and 29.76 ± 5.54 years in the overweight/obese group (*p* = 0.554).

The age distribution in the overall cohort (*n* = 65) revealed that the highest proportion of pregnancies occurred in women aged 25–29 years (36.0%), followed by the 30–34 year group (26.7%), and 35–39 year group (18.7%). Only 4.0% of pregnancies occurred in adolescents aged 15–19 years, and no pregnancies were recorded in women aged 40 years or older. The distribution of maternal age did not differ significantly between BMI-defined subgroups.

The mean number of previous pregnancies was slightly higher in the normal BMI group (1.88 ± 0.97) than in the overweight/obese group (1.58 ± 0.96), although this difference did not reach statistical significance (*p* = 0.124). Parity levels were also similar across groups, with no significant differences observed. The average time since liver transplantation was 6.65 ± 4.50 years in the normal BMI group and 7.76 ± 5.58 years in the overweight/obese group (*p* = 0.401).

### 3.2. Liver Transplantation Characteristics

#### 3.2.1. Indications for Transplantation

The underlying liver diseases leading to transplantation were diverse and reflected the typical spectrum of chronic liver failure in reproductive-aged women. The most frequent indication was Wilson’s disease (*n* = 17, 23.6%), a genetic disorder of copper metabolism prevalent among younger individuals, followed by autoimmune hepatitis (AIH; *n* = 12, 16.7%) and viral hepatitis (*n* = 10, 13.9%), primarily hepatitis B or C. Primary sclerosing cholangitis (PSC) was also observed as an indication in eight events (11.1%), consistent with its autoimmune association and prevalence in younger adults.

Other etiologies included biliary atresia (*n* = 7, 9.7%) and idiopathic liver failure (*n* = 6, 8.3%). Less frequent causes included Budd–Chiari syndrome (*n* = 4), toxic liver injury (*n* = 4), and parasitic infections (*n* = 4), which, although rare in high-income countries, were identified in a minority of cases. Single cases of liver malignancy (*n* = 2) and liver hemangioma (*n* = 1) were also reported.

The distribution of indications was broadly similar between the BMI subgroups [Table 1], with Wilson’s disease and viral hepatitis evenly distributed, while autoimmune hepatitis appeared slightly more frequently among women with an elevated BMI. However, these differences did not reach statistical significance (*p* = 0.528), indicating comparable underlying disease profiles regardless of BMI status.

#### 3.2.2. Immunosuppressive Regimens

All patients received long-term immunosuppressive therapy during pregnancy. The most commonly administered agent was tacrolimus, used in 62 cases (86.1%), consistent with its favorable safety profile in pregnancy. Corticosteroids were prescribed in 50 events (69.4%), often as part of a dual or triple therapy protocol. Cyclosporine was used in 11 events (15.3%) and azathioprine in 8 events (11.1%).

There were no statistically significant differences in the use of these agents across BMI categories [Table 2]. The immunosuppressive regimens appeared to be selected primarily based on transplant status and prior drug tolerance, rather than body weight considerations.

#### 3.2.3. Pre-Pregnancy Comorbidities

In addition to transplant-related conditions, several cases in the cohort had pre-existing chronic diseases prior to conception. Chronic hepatitis C virus (HCV) infection was documented in five cases (6.9%), likely reflecting previous exposure and viral persistence post-transplant. Arterial hypertension diagnosed before pregnancy was present in four cases (5.6%), and pregestational diabetes mellitus was recorded in two cases (2.8%). These comorbidities were distributed across both BMI groups without significant differences, suggesting that an elevated BMI was not specifically associated with a higher baseline burden of chronic disease in this transplant population.

#### 3.2.4. Time Interval Between Transplantation and Pregnancy

The interval between liver transplantation and conception varied across the cohort, though most pregnancies occurred more than three years post-transplantation, aligning with current clinical recommendations. Specifically, 34.7% of events occurred 5–10 years after transplantation, 27.8% after more than 10 years, and 27.8% between 3 and 5 years. Only 13.9% of conceptions took place within the first three years post-transplant, suggesting cautious timing in most cases [Table 3].

There were no statistically significant differences between BMI subgroups with regard to the timing of pregnancy in relation to transplantation (*p* = 0.336). This suggests that BMI status did not influence the clinical decision regarding the optimal time for conception following transplant surgery.

### 3.3. Pregnancy Outcomes

#### 3.3.1. Obstetric Outcomes

Gestational age at delivery did not significantly differ between groups. The normal BMI group had a mean gestational age of 37.16 ± 1.80 weeks, whereas the overweight/obese group delivered at a mean of 37.02 ± 2.12 weeks (*p* = 0.995), indicating that the majority of pregnancies reached term in both cohorts. Preterm birth (<37 weeks) was not significantly more frequent in either group.

Cesarean section was the most frequent mode of delivery overall, occurring in 69.4% (*n* = 50) of all cases. Cesarean delivery was more common in the overweight/obese group (76.0%) compared to the normal BMI group (64.6%), though the difference was not statistically significant (*p* = 0.465). Vaginal delivery occurred in 25.0% of overweight/obese pregnancies and 35.4% of those with normal BMI.

#### 3.3.2. Neonatal Outcomes

The neonatal condition at birth was consistently favorable across both groups. The mean 5 min Apgar scores were 9.71 ± 0.65 for the normal BMI group and 9.83 ± 0.48 for the overweight/obese group (*p* = 0.396), with no scores below 7 observed in either group.

The average neonatal birth weight was 3043.4 ± 612.7 g in the normal BMI group and 2924.7 ± 604.7 g in the overweight/obese group (*p* = 0.792). Birth length averaged 52.27 ± 3.90 cm and 53.00 ± 3.54 cm in the respective groups. When adjusted for gestational age, the average birth weight percentile was 51.48 ± 29.53 in the normal BMI group versus 44.85 ± 33.52 in the overweight/obese group (*p* = 0.542).

#### 3.3.3. Pregnancy Complications

Pregnancy-related complications were reported in 64.6% of events in the normal BMI group and in 76.0% of those in the overweight/obese group (*p* = 0.465). The most frequent complication was urinary tract infection (UTI), diagnosed in 31.25% of normal BMI pregnancies and 40.0% of those with an elevated BMI (*p* = 0.626).

Maternal anemia occurred in 35.42% of normal BMI events and 32.0% of those with an elevated BMI (*p* = 0.974). Pregnancy-induced hypertension (PIH) was recorded in 10.42% of normal BMI cases and 28.0% of overweight/obese cases (*p* = 0.112). Postpartum hemorrhage (PPH) occurred in 16.67% of the normal BMI group and in 8.0% of the overweight/obese group (*p* = 0.507).

Gestational diabetes mellitus (GDM) was diagnosed in 6.25% of the normal BMI group and 4.0% of the overweight/obese group (*p* = 0.888). No statistically significant differences were found in the incidence of any individual pregnancy complication between the BMI groups.

A complete summary of clinical characteristics and comparisons between the BMI subgroups is presented in Table 4.

## 4. Discussion

### 4.1. Main Findings

In this retrospective analysis of liver transplant recipients, the authors found no statistically significant association between pre-pregnancy BMI and key maternal or perinatal outcomes. Despite numerically higher rates of pregnancy complications such as PIH and cesarean delivery among overweight/obese women, these differences did not reach statistical significance. Similarly, gestational age at delivery, birth weight, and Apgar scores were comparable across BMI categories, with both groups achieving favorable perinatal results. These results may indicate that, within the highly controlled clinical environment of liver transplant care—characterized by close monitoring and individualized management—the impact of elevated BMI on pregnancy outcomes may be less pronounced than in the general obstetric population. Another possible explanation is that the small sample size limited the statistical power to reveal the true differences between BMI groups.

### 4.2. BMI Profile Compared to General Population

A notable contextual finding of our study is the differing pre-pregnancy BMI distribution in liver transplant (LTx) recipients compared to the general population of Polish women of reproductive age. In our cohort, 64.0% had a normal BMI and only 6.9% were obese, while national data report obesity rates of 15–18% and a lower prevalence of normal weight (50–55%) among women aged 18–44 [17,18]. Several factors may explain this difference. Transplant recipients typically undergo thorough preconception counseling, where modifiable risks like excess weight are addressed. Additionally, chronic liver disease and immunosuppression may affect metabolism and body composition [19]. A degree of selection bias is also likely, as women with severe obesity or poorer health may be discouraged from conceiving or have reduced fertility [20]. This is because an elevated BMI is recognized as a key factor influencing pregnancy outcomes. Maternal obesity has been linked to an increased likelihood of various complications, including gestational diabetes, pregnancy-induced hypertension, preeclampsia, venous thromboembolism, and a higher rate of cesarean delivery [21]. Importantly, the relatively small sample size in our study limits the ability to detect significant differences between BMI categories. These observations align with international data, where BMI optimization is emphasized in reproductive planning post-transplant. Maintaining normal BMI has been associated with improved maternal and neonatal outcomes and reduced graft-related risks [22,23,24]. Therefore, the relatively favorable obstetric outcomes observed in our cohort may, at least in part, be attributed to the optimized BMI profile of the study population. The markedly lower prevalence of obesity compared to the general population could have attenuated the expected impact of elevated BMI on adverse pregnancy outcomes. These findings underscore the importance of individualized, multidisciplinary preconception care in optimizing maternal health among liver transplant recipients.

### 4.3. Role of Liver Disease Etiology and Metabolic Context

When discussing the relevance of pre-pregnancy BMI in relation to obstetric outcomes among liver transplant recipients, it is important to take into account the underlying liver disease that necessitated the transplantation. The primary condition itself may predispose patients to metabolic disturbances or obesity—particularly in cases such as nonalcoholic steatohepatitis (NASH)—and can independently influence pregnancy outcomes [25]. For instance, NASH is strongly associated with the features of metabolic syndrome, including insulin resistance and dyslipidemia, which may affect both maternal physiology and fetal development [25]. Additionally, other chronic liver diseases such as autoimmune hepatitis or Wilson’s disease may contribute to systemic inflammation or endocrine dysregulation, further complicating pregnancy even before considering BMI [26,27]. Therefore, the baseline hepatic pathology must be viewed as a potential confounder when interpreting the relationship between BMI and obstetric outcomes in this high-risk population.

### 4.4. Impact of Immunosuppressive Therapy

Could modifications in immunosuppressive therapy before and during pregnancy influence the metabolic profile, thereby potentially affecting obstetric outcomes? While data are limited, some studies suggest that immunosuppressive agents can impact lipid metabolism. For instance, a study comparing azathioprine and mycophenolate mofetil (MMF) in renal transplant patients found that total cholesterol and triglyceride levels significantly increased in both groups during the first year post-transplantation. Notably, at the three-month mark, total cholesterol levels were higher in the MMF group compared to the azathioprine group, although no significant differences were observed at later time points [28]. These findings indicate that MMF may transiently exacerbate dyslipidemia, which could have implications for pregnancy outcomes, especially in populations already at risk of metabolic complications.

Given that MMF is contraindicated during pregnancy due to its teratogenicity, patients are often transitioned to alternative immunosuppressive regimens prior to conception [11]. This switch could potentially alter the metabolic profile, but the specific effects of such changes on pregnancy outcomes remain underexplored. Further research is needed to elucidate the impact of immunosuppressive therapy modifications on metabolic parameters and their subsequent influence on obstetric outcomes.

In our study, however, we did not observe significant differences in the use of immunosuppressive therapy between women with normal BMI and those with an elevated BMI, as shown in Table 2. The most commonly used drugs, including steroids, tacrolimus, cyclosporine, and azathioprine, were similarly distributed across both groups, with no statistically significant variation in their use (*p* > 0.05 for all comparisons), suggesting that the modification of immunosuppressive regimens prior to pregnancy may not have a significant impact on metabolic profiles in this cohort. Despite this, the potential for immunosuppressive therapy to influence metabolic parameters such as lipid levels and glucose metabolism remains an important area for further research, particularly in relation to pregnancy outcomes. Understanding how specific immunosuppressive agents affect maternal metabolic processes could help improve preconception counseling and lead to more individualized care for liver transplant recipients planning pregnancy.

### 4.5. Methodological Considerations

One potential explanation for the lack of significant associations between pre-pregnancy BMI and adverse obstetric outcomes in our cohort may relate to the BMI threshold applied in the analysis. The commonly used cutoff of 25 kg/m^2^ to define overweight might not adequately capture the severity of metabolic dysregulation associated with higher levels of adiposity. It is possible that the impact of elevated BMI becomes more pronounced only at higher thresholds, such as BMI ≥ 30 kg/m^2^, which corresponds to clinical obesity.

### 4.6. Strengths of the Study

This study offers several distinct strengths that enhance its scientific and clinical value. It focuses on a highly specific and medically complex population—pregnant liver transplant recipients—who remain significantly underrepresented in the existing obstetric and transplant literature. By investigating this unique cohort, the study contributes to a better understanding of maternal and neonatal risks in the setting of solid organ transplantation. The extended study period covering over two decades provides a robust longitudinal dataset that captures evolving trends in clinical practice, immunosuppressive therapy, and perinatal care. Such temporal breadth increases the relevance and external validity of the findings, particularly in the context of contemporary transplant medicine. The single-center design ensures uniformity in clinical management, data recording, and obstetric protocols, thereby minimizing inter-institutional variability. Conducted at a tertiary care academic center with extensive experience in both liver transplantation and high-risk pregnancy management, the study benefits from a high degree of diagnostic and therapeutic consistency. A further strength lies in the use of internationally standardized BMI categories to stratify patients prior to conception. This enables a focused examination of a modifiable metabolic parameter that is highly relevant to current guidelines on maternal health and preconception care. It also aligns the study with broader global efforts to assess obesity-related risks in pregnancy. Importantly, the analysis encompasses a comprehensive range of obstetric and neonatal endpoints, including hypertensive disorders, mode of delivery, gestational age at birth, neonatal weight, Apgar scores, and preterm delivery rates. Such a multidimensional outcome profile allows for a more nuanced interpretation of the risk associated with an elevated BMI in transplant recipients.

This study also adds regional novelty to the literature, potentially representing one of the first investigations of its kind within Central or Eastern Europe. This geographic specificity enhances its contribution to the global understanding of post-transplant reproductive health. The clinical implications are considerable, as the findings support the integration of BMI assessment and weight optimization strategies into individualized preconception counseling and risk stratification for transplant recipients planning pregnancy.

### 4.7. Limitations of the Study

Although the present study has notable strengths, several limitations should be acknowledged when interpreting the findings. The retrospective nature of the analysis introduces inherent constraints related to data completeness, potential selection bias, and the inability to control for all relevant confounding variables. As data were collected from existing medical records over a 21-year period, variations in documentation standards and clinical practices across time may have affected data uniformity and quality. The single-center design, while beneficial for consistency, may limit the generalizability of the results to broader populations, particularly given institutional differences in transplant protocols, immunosuppressive regimens, and obstetric management.

Moreover, the relatively small sample size—although comparable to other studies in this rare patient population—reduces statistical power and may have limited the detection of significant differences, especially for less frequent outcomes such as preeclampsia or neonatal complications. BMI was the primary metabolic parameter examined; however, other potentially relevant factors such as gestational weight gain [29], nutritional status, physical activity, or markers of metabolic syndrome were not assessed due to data availability.

Additionally, while patients were stratified by BMI using standard thresholds, the cross-sectional nature of BMI measurement does not capture the fluctuations in weight or body composition that may have occurred during the preconception or gestational period [30]. Information on immunosuppressive drug levels, changes in dosage during pregnancy, and their potential interaction with metabolic outcomes was limited and not systematically analyzed [31,32]. These factors could play a significant role in mediating the relationship between pre-pregnancy BMI and pregnancy outcomes in liver transplant recipients. Finally, the heterogeneity in the etiology of underlying liver disease, graft function, and comorbidities was not fully accounted for in subgroup analyses. These variables may influence both the baseline metabolic risk and pregnancy-related complications, and their interaction with BMI warrants further investigation in future prospective studies. Additionally, although the study combines overweight and obese women into a single group, it is important to acknowledge that these categories may represent distinct physiological profiles. Merging them may obscure any potential differences that could influence the results. Therefore, future studies could benefit from BMI stratification (e.g., 25–29.9 vs. ≥30) to better capture the heterogeneity within these groups.”

In the analyzed cohort, the proportion of patients meeting the criteria for obesity may have been too small to detect statistically significant differences, especially in the context of a relatively rare and medically complex population such as liver transplant recipients. Furthermore, individuals with overweight but not obesity may not exhibit the same degree of metabolic disturbances, such as insulin resistance, dyslipidemia, or inflammation, that typically mediate adverse pregnancy outcomes. As such, future studies might benefit from stratifying patients by finer BMI categories or analyzing BMI as a continuous variable to better reflect the dose-dependent nature of obesity-related risk.

## 5. Conclusions

In the presented single-center cohort of liver transplant recipients, pre-pregnancy BMI did not show a statistically significant association with adverse maternal or neonatal outcomes. Although rates of certain complications were numerically higher among overweight and obese women, the overall obstetric outcomes remained favorable across BMI subgroups. These findings suggest that, within the setting of specialized transplant care and structured preconception counseling, elevated BMI may have a less pronounced impact on pregnancy outcomes compared to the general obstetric population.

Nevertheless, limitations related to sample size, retrospective design, and the lack of extended metabolic profiling highlight the need for future prospective studies. Such research should include larger multicenter cohorts, detailed assessments of metabolic health (e.g., gestational weight gain, insulin resistance, lipid profiles), and the longitudinal monitoring of immunosuppressive therapies. Applying finer BMI stratification and incorporating additional nutritional and cardiovascular markers may further improve individualized risk assessment. Overall, these efforts may enhance the safety and success of pregnancy in women with solid organ transplants.

## Figures and Tables

**Table 1 diagnostics-15-02054-t001:** Indications for liver transplantation.

**Indication**	**Total**	**BMI 18.5–24.9**	**BMI ≥ 25**	Chi^2^ = 9.034 *p* = 0.528
Wilson’s disease	17	11	5	
Autoimmune hepatitis (AIH)	12	4	7	
Viral hepatitis	10	5	5	
Primary sclerosing cholangitis (PSC)	8	6	2	
Biliary atresia	7	4	3	
Idiopathic	6	3	3	
Budd–Chiari syndrome	4	3	1	
Toxic liver injury	4	2	2	
Parasitic infection	4	0	4	
Liver malignancy	2	2	0	
Liver hemangioma	1	1	0	

**Table 2 diagnostics-15-02054-t002:** Immunosuppressive therapy.

Drug	Total	BMI 18.5–24.9	BMI ≥ 25	
Steroids	50	33	17	Chi^2^_Yates_ = 0.040 *p* = 0.841
Tacrolimus	62	41	21	Chi^2^_Yates_ = 0.034 *p* = 0.854
Cyclosporine	11	7	4	Chi^2^_Yates_ = 0.034 *p* = 0.854
Azatioprine	8	4	4	Chi^2^_Yates_ = 0.360 *p* = 0.548

‘Yates’ correction for continuity: A correction applied to the chi-square test in 2 × 2 contingency tables when sample sizes are small. This adjustment helps improve the accuracy of the test by addressing the continuity correction in the approximation of expected frequencies.

**Table 3 diagnostics-15-02054-t003:** Period between liver transplantation and delivery.

**Years Since** **Transplantation**	**Total**	**BMI 18.5–24.9**	**BMI ≥ 25**	Chi^2^ = 3.388 *p* = 0.336
1–3 years	10	6	4	
3–5 years	20	12	8	
5–10 years	25	10	14	
>10 years	20	13	6	

**Table 4 diagnostics-15-02054-t004:** Differences in continuous variables between women with normal pre-pregnancy BMI and those who were overweight or obese.

Variable	Group	Mean (M)	SD	Min	Q1	Median (Me)	Q3	Max	Test	*p*-Value
Maternal age at delivery (years)	I	28.96	5.42	15	26.5	28	32	39	t = −0.595	0.554
	II	29.76	5.54	18	26	32	34	37		
Years since transplantation	I	6.65	4.50	1	3	5.5	10	17	Z = −0.840	0.401
	II	7.76	5.58	1	4	6	9	21		
Pre-pregnancy BMI (kg/m^2^)	I	27.90	2.70	25.00	26.33	27.36	29.07	37.73	Z = −6.970	< 0.001
	II	22.51	1.75	18.73	21.47	22.93	23.90	24.91		
Age at transplantation (years)	I	22.00	8.79	1	17	24	30	36	Z = −0.186	0.852
	II	22.31	5.93	6	19	24	26.50	33		
Gestational week at delivery	I	37.16	1.80	33	36	37	38	41	Z = 0.006	0.995
	II	37.02	2.12	32	35.5	37.5	39	41		
Apgar score at 5 min	I	9.83	0.48	8	10	10	10	10	Z = −0.848	0.396
	II	9.71	0.65	7	10	10	10	10		
Birth weight (g)	I	3043.4	612.7	1640	2780	3150	3520	4100	t = −0.792	0.431
	II	2924.7	604.7	1800	2537.5	2935	3270	4300		
Percentile	I	51.48	29.53	2	37	58	74	97	Z = −0.611	0.542
	II	44.85	33.52	3	14	37	80	99		

Abbreviations: I—normal BMI group; II—overweight or obese group; M—mean; SD—standard deviation; Min—minimum; Q1—first quartile; Me—median; Q3—third quartile; Max—maximum; Z—Mann–Whitney U test; t—Student’s *t*-test; *p*—significance level.

## Data Availability

Data are available from the corresponding author upon reasonable request.

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
