# Peer review of "Impact of Pre-Pregnancy Body Mass Index on Pregnancy and Perinatal Outcomes in Liver Transplant Recipients: A Retrospective Cohort Study"

_diagnostics, 2025, doi:10.3390/diagnostics15162054_

Round 1

Reviewer 1 Report

Comments and Suggestions for Authors

Authors present a retrospective cohort study investigating the association between pre-pregnancy Body Mass Index (BMI) and pregnancy/perinatal outcomes in female liver transplant recipients. The study is well-designed, the methodology is clearly described, and the results are clearly presented.

However, Authors need to address some minor points before the manuscript can be suitable for publication:

  • Clarity on "Trends":The abstract mentions "observed trends suggest a potential predisposition to hypertensive disorders." I suggest to insert the statistical data and p values.
  • The abstract states "73 pregnancies included in the analysis," but the Results section later clarifies that 72 deliveries were ultimately analyzed after excluding the underweight group.  Please check and revise
  • Please state clearly at the end of the Introduction section the principal aim of the study
  • The authors state (lines 129-130) that each delivery was treated as an independent case and referred to as a "patient." It coud be confusing to the reader for patients with multiple pregnancies, so I suggest that each delivery would be referred as “event”
  • In the Methods please state the specific software version used .
  • Table 2: please define in a footnote what "Yates" refers to (Yates' correction for continuity), for readers less familiar with the test.

Author Response

Comment 1:

„The abstract mentions "observed trends suggest a potential predisposition to hypertensive disorders". I suggest to insert the statistical data and p values.”

Response 1:

Thank you for your suggestion. We have updated the abstract by including the statistical data and p-values as follows: „Nevertheless, the observed trends suggest a potential predisposition to hypertensive disorders (PIH: 28% vs. 10.4%, p = 0.112), underscoring the importance of individualized preconception counseling and weight optimization strategies in this this high-risk patient population.” (Abstract, page 1, line 28)

Comment 2:

„The abstract states "73 pregnancies included in the analysis", but the Results section later clarifies that 72 deliveries were ultimately analyzed after excluding the underweight group. Please check and revise.”

Response 2:

We apologize for the mistake and thank you for pointing it out. The "Results" section correctly states that 72 deliveries were ultimately analyzed after excluding the underweight group. We have now revised the abstract to ensure consistency, reflecting the correct total number of pregnancies included in the study, as well as the percentages of women in the normal BMI and elevated BMI groups as follows: „Among 72 pregnancies included in the analysis, 48 (66.7%) were in women with normal BMI, and 24 (33.3%) in those with elevated BMI.” (Abstract, page 1, lines 19-20)

Comment 3:

„Please state clearly at the end of the introduction section the principal aim of the study.”

Response 3:

Thank you for your suggestion. We have revised the end of the introduction to clearly state the principal aim of the study. The updated sentence now reads: „The principal aim of this study was to evaluate the association between pre-pregnancy body mass index (BMI) and pregnancy-related complications and neonatal outcomes in female liver transplant recipients.” (Introduction, page 3, lines 111-113)

Comment 4:

„The authors state (lines 129-130) that each delivery was treated as an independent case and referred to as a "patient". It could be confusing to the reader for patients with multiple pregnancies, so I suggest that each delivery would be referred as event.”

Response 4:

We appreciate the reviewer’s comment. We revised the terminology and standardized the terminology – each case was refereed as “event” in the all result section in the manuscript.

“Each delivery was treated as an independent case and referred to as an 'event' in the results section, in order to minimize potential misinterpretations and analytical errors.” (Page 3, line 138)

Comment 5:

„In the Methods, please state the specific software version used.”

Response 5:

We appreciate the reviewer’s comment. The programme used to the statistical analysis was Statistica version 14.2.0. We updated this information in the article: „Statistical analysis was performed using Statistica version 14.2.0 programme.”  (page 3, line 140)

Comment 6:

„Table 2: please define in a footnote what "Yates" refers to (Yates’ correction for continuity), for readers less familiar with the test.”

Response 6:

Thank you for your suggestion. We have added a footnote to Table 2 to clarify what "Yates" refers to. The updated table now includes the following footnote: „Yates’ correction for continuity: A correction applied to the chi-square test in 2x2 contingency tables when sample sizes are small. This adjustment helps improve the accuracy of the test by addressing the continuity correction in the approximation of expected frequencies.”  (Table 2, page 5, lines 213-215)

Reviewer 2 Report

Comments and Suggestions for Authors

This clinically significant, retrospective, single-center cohort study addresses a specific and rarely investigated population—pregnant women after liver transplantation—and examines the impact of pre-pregnancy body mass index (BMI) on pregnancy outcomes. It represents an important contribution and sheds light on the anthropometric and clinical characteristics of this population in Poland.

The introduction is well written and informative. The study is well designed, and considering the rarity of the population, the retrospective design is appropriate for this research question. The inclusion and exclusion criteria are clearly defined. The decision to exclude the underweight group from analysis due to a small sample size is justified.

However, merging overweight and obese women into a single group may mask potential differences. Therefore, it would be advisable for the authors to either perform further BMI stratification (e.g., 25–29.9 vs. ≥30) or at least acknowledge this limitation in the discussion.

The statistical tests used in the analysis are appropriate. The tables, however, are somewhat extensive and not easily readable; I suggest the authors revise and improve their presentation. Additionally, the text refers to “Graph 1,” which is not included in the manuscript—please ensure it is added.

There is also inconsistency in terminology, with the alternating use of “patients” and “deliveries”; I recommend the authors standardize this throughout the manuscript.

The results are clearly presented, and the lack of statistically significant differences is appropriately and cautiously interpreted. The discussion could be expanded with respect to immunosuppressive therapy. The conclusion is concise and consistent with the findings. 

Author Response

Comment 1:

„However, merging overweight and obese women into a single group may mask potential differences. Therefore, it would be advisable for the authors to either perform further BMI stratification (e.g., 25-29.9 vs. ≥30) or at least acknowledge this limitation in the discussion.”

Response 1:

We appreciate the reviewer’s comment. We acknowledge that combining overweight and obese women into one group may obscure potential differences. To address this, we have added the following statement in the limitations section of the discussion: „Additionally, although the study combines overweight and obese women into a single group, it is important to acknowledge that these categories may represent distinct physiological profiles. Merging them may obscure potential differences that could influence the results. Therefore, future studies could benefit from BMI stratification (e.g., 25-29.9 vs. ≥30) to better capture the heterogeneity within these groups.” (Discussion, 4.7. Limitations of the Study, page 11, lines 441-445)

Comment 2:

„The tables, however, are somewhat extensive and not easily readable; I suggest the authors revise and improve their presentation.”

Response 2:

Thank you for your suggestion. We have reviewed the tables and retained only the essential information in order to enhance their clarity for the reader.

Comment 3:

„Additionally, the text refers to "Graph 1", which is not included in the manuscript. Please ensure it is added.”

Response 3:

We apologize for the confusion. There is no "Graph 1" in the manuscript, and this reference was a mistake. All the information that was intended to be conveyed through the graph is thoroughly described in the subsequent sentences of the manuscript.

Comment 4:

There is also inconsistency in terminology with the alternating use of "patient" and "deliveries"; I recommend the authors standardize this without the manuscript.

Response 4:

We appreciate the reviewer’s comment. We revised the terminology and standardized the terminology – each case was refereed as “event” in the all result section in the manuscript. “Each delivery was treated as an independent case and referred to as an 'event' in the results section, in order to minimize potential misinterpretations and analytical errors.” (Page 3, line 138)

Comment 5:

„The discussion could be expanded with respect to immunosuppressive therapy.”

Response 5:

We appreciate the reviewer’s comment. To address this, we have added the following statement: „In our study, however, we did not observe significant differences in the use of immunosuppressive therapy between women with normal BMI and those with elevated BMI, as shown in Table 2. The most commonly used drugs, including steroids, tacrolimus, cyclosporine, and azathioprine, were similarly distributed across both groups, with no statistically significant variation in their use (p > 0.05 for all comparisons), suggesting that the modification of immunosuppressive regimens prior to pregnancy may not have a significant impact on metabolic profiles in this cohort. Despite this, the potential for immunosuppressive therapy to influence metabolic parameters such as lipid levels and glucose metabolism remains an important area for further research, particularly in relation to pregnancy outcomes. Understanding how specific immunosuppressive agents affect maternal metabolic processes could help improve preconception counseling and lead to more individualized care for liver transplant recipients planning pregnancy.”  (Discussion, 4.4. Impact of Immunosuppressive Therapy, page 9-10, lines 360-371)
